# R4: Nested Reasoning-Retrieval for Reward Modeling in Role-Playing Agents

**Renzhi Wang**[1*]**, Chongqiang Wei**[2]**, Zhisheng Wang**[2]**, Piji Li**[1†]
[1]Nanjing University of Aeronautics and Astronautics [2]Tencent
{rzhwang,pjli}@nuaa.edu.cn

## Abstract

Role-playing dialogue presents unique challenges for large language models (LLMs): beyond producing coherent text, models must sustain character persona, integrate contextual knowledge, and convey emotional nuance. Despite strong reasoning abilities, current LLMs often generate dialogue that is literal, stylistically bland, and misaligned with character-specific traits. Existing approaches such as retrieval-augmented generation (RAG) or reinforcement learning (RL) with scalar rewards are insufficient, as they cannot capture nuanced preferences or adapt reliably to diverse character contexts. In this work, we introduce R4, a unified framework that equips both the reward model and the role-playing agent with reasoning and retrieval capabilities. Our reward model reformulates evaluation as structured reasoning: it integrates multi-step deliberation and retrieved knowledge to assess responses along multiple dimensions. This reward supervision is then used within reinforcement learning to train a dialogue agent with the same dual capabilities, enabling contextually grounded and persona-consistent generation. Experiments demonstrate that R4 substantially improves dialogue quality, particularly in persona fidelity, narrative coherence, and emotional expressiveness. Analysis of training dynamics and case studies further shows that R4 agents employ retrieval more effectively, engage in retrieval-informed self-reflection, and achieve emergent role-playing behaviors unattainable by prior methods.

## 1 Introduction

Large language models (LLMs), such as DeepSeek-R1 (DeepSeek-AI et al., 2025) and OpenAI's o series (OpenAI, 2024), have demonstrated remarkable reasoning capabilities across diverse tasks, including multi-hop question answering, code generation, and mathematical problem-solving. However, this reasoning prowess does not readily translate to open-domain dialogue, particularly in role-playing scenarios that require embodying established characters from rich narrative sources like novels. In this context, success is not defined by factual accuracy alone, but by a nuanced fusion of character coherence, factual consistency, and emotional engagement (Feng et al., 2025). Responses generated by reasoning models are often overly literal, stylistically flat, and insufficiently grounded in the speaker's persona, undermining the potential for immersive interaction.

The challenge lies in a fundamental misalignment between the objectives of conventional LLMs and the demands of authentic role-playing. Specifically: (1) reasoning-oriented models favors correctness, which leads to formulaic and rigid dialogue, sacrificing the expressiveness vital for role-playing, and (2) the absence of role-specific knowledge—such as backstory, relationships, and evolving motivations—undermines narrative continuity and realism.

Retrieval Augmented Generation (RAG) has emerged as a promising approach for incorporating external knowledge into LLM outputs. While beneficial in knowledge-centric tasks, standard one-shot RAG approaches prove inadequate for role-playing dialogue due to static query formulation and inability to adapt to evolving conversational contexts. Advanced retrieval systems, such as DeepResearcher (Zheng et al., 2025) and WebThinker (Li et al., 2025b), offer more sophisticated

---

*Work done during internship in Tencent.
†Corresponding author.

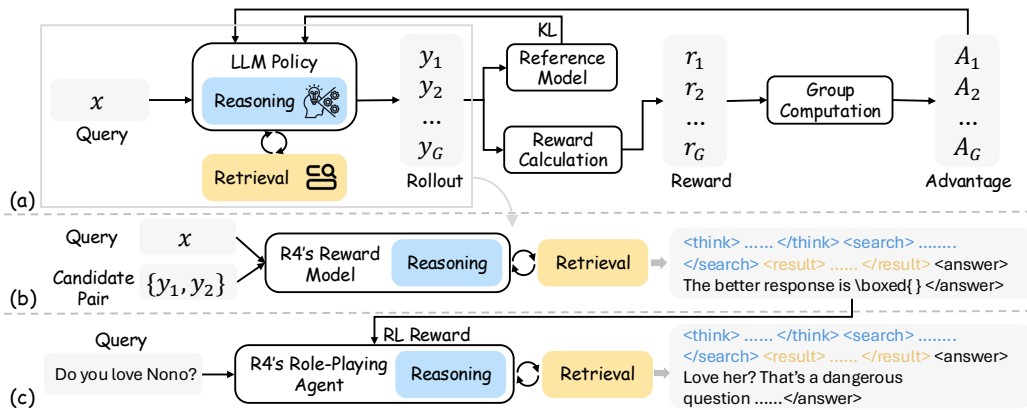

Figure 1: Architecture of R4 framework. (a) Training pipeline. (b-c) Reward model performs structured reasoning over response pairs and assigns comparative rewards to guide advantage estimation and policy updates. Both the agent and reward model integrate reasoning and retrieval throughout.

retrieval dynamics but rely heavily on handcrafted prompts and rule-based orchestration, making them brittle and computationally expensive. Recent iterative search frameworks like ReSearch (Chen et al., 2025a) and Search-R1 (Jin et al., 2025) integrate retrieval more seamlessly through reinforcement learning, yet still underperform in role-playing contexts where success depends not only on factual grounding but also on persona alignment and emotional authenticity.

A key challenge in applying reinforcement learning to role-playing dialogue is constructing reliable, multi-dimensional reward signals that capture the subjective and multifaceted nature of character portrayal. Unlike structured tasks such as code generation or mathematical reasoning—where correctness is objectively measurable—role playing dialogue quality encompasses persona consistency, emotional appropriateness, narrative coherence, and stylistic authenticity. As noted by (Tu et al., 2024), evaluating candidate responses requires interpreting user intent, identifying appropriate evaluation criteria, and reasoning over implicit character motivations and narrative cues. This complexity necessitates reward models capable of structured reasoning grounded in character-specific knowledge.

However, current reward modeling approaches fall short. Our systematic analysis across diverse characters and evaluation contexts reveals two fundamental biases: (1) **role bias**: evaluation consistency varies dramatically based on character familiarity, with well-known characters receiving more stable assessments while lesser-known characters suffer from inconsistent scoring; and (2) **reference bias**: the availability of character-specific contextual information significantly impacts evaluation quality, with models producing more accurate and consistent judgments when provided with relevant background knowledge. These findings reveal a critical gap: existing reward models evaluate responses in isolation, without the contextual reasoning and character-specific grounding essential for reliable supervision in role-playing scenarios. Consequently, reward models, like dialogue agents themselves, need to integrate reasoning and retrieval to generate credible supervision signals.

To address these challenges, we propose **R4**, a unified framework that endows both the **R**eward model and the **R**ole-playing dialogue agent with the ability to **R**eason and **R**etrieve. Our approach consists of three core components: (1) a **character-specific knowledge construction** pipeline that systematically extracts and organizes persona-relevant information from narrative sources, particularly focusing on literary characters from novels; (2) a **reasoning-augmented reward model** that performs structured multi-dimensional evaluation through reasoning chains grounded in retrieved character context; and (3) a **role-playing agent** that integrates the same dual capabilities to generate contextually appropriate and character-consistent responses. The main contributions of this work are:

- We reformulate reward modeling as a structured reasoning task, introducing a novel reward model architecture that integrates multi-dimensional evaluation through reasoning chains and retrieved knowledge, systematically addressing role bias and reference bias in existing approaches.

- We propose an end-to-end training framework that unifies reasoning and retrieval across reward modeling and response generation, enabling more reliable supervision and higher-quality dialogue tailored to literary character role-playing.

- We show across model scales that our approach consistently enhances character fidelity, emotional expressiveness, and overall dialogue quality over existing methods.

## 2 METHODOLOGY: R4

Existing methods often treat reward modeling and response generation as disjoint components, limiting their ability to reason contextually and adapt to character-specific knowledge. In contrast, our proposed R4 framework tightly integrates reasoning and retrieval mechanisms across both the reward model and the dialogue agent. In what follows, we describe the full design of R4.

### 2.1 CHARACTER-SPECIFIC KNOWLEDGE CONSTRUCTION

To enable effective retrieval-augmented reasoning for role-playing dialogue, we develop a comprehensive knowledge construction pipeline that systematically extracts, organizes, and maintains character-specific contextual information from narrative sources. This knowledge repository serves as foundational external memory accessed by both reward model and dialogue agent throughout training and inference, ensuring consistent character grounding across all system components.

**Narrative Segmentation and Analysis**. Our pipeline begins by processing narrative texts through an LLM-based segmentation strategy that partitions content into semantically coherent and plot-relevant units, inspired by (Duarte et al., 2024). For each identified segment, we employ structured prompting to generate comprehensive character-centric representations that capture multiple dimensions of character information: (1) **persona traits** including personality characteristics, behavioral patterns, and distinctive mannerisms; (2) **emotional states** reflecting both explicit emotional expressions and latent psychological conditions; (3) **contextual knowledge** encompassing character backgrounds, relationships, and domain-specific expertise; and (4) **narrative goals** representing both short-term objectives and long-term character arcs within the story context.

**Knowledge Organization and Indexing**. The extracted character information is structured into a hierarchical knowledge organization that supports efficient retrieval during reasoning processes. Each knowledge entry is associated with multiple indexing keys, including character identifiers, emotional contexts, relationship dynamics, and narrative situations. This multi-faceted indexing enables precise retrieval of relevant information based on conversational context and character interaction patterns. Additionally, we implement semantic clustering to group related knowledge entries, facilitating multi-hop reasoning scenarios where the model needs to connect disparate pieces of character information.

**Dynamic Knowledge Expansion and Validation**. To enhance coverage and adapt to behaviors observed during model training, we implement a dynamic expansion mechanism that continuously enriches the knowledge base. During both reward model and dialogue agent training, we collect retrieval queries generated by the models and use them to identify knowledge gaps. These queries are then processed through automated synthesis using advanced language models (e.g., GPT-4o) and human-authored annotations for critical aspects requiring nuanced understanding. To ensure reliability, we employ automated consistency checks and periodic expert reviews to maintain knowledge quality and coherence across different narrative contexts.

Implementation details are provided in Appendix B. This character-specific knowledge construction pipeline forms the backbone of the R4 framework, providing both the reward model and dialogue agent with access to rich, contextually relevant information necessary for generating authentic and engaging role-playing interactions.

### 2.2 REWARD MODEL

We now introduce the reasoning- and retrieval-augmented reward model in R4, designed to address fundamental limitations in conventional reward modeling approaches for role-playing dialogue.

**Biases in Existing Reward Models.** Current reward modeling approaches fall into two main paradigms. **Scalar-based reward models** (ScalarRM) (Liu et al., 2024a) formulate evaluation as binary or ordinal classification tasks, offering computational efficiency but providing opaque reward signals with no interpretable reasoning process. **Generative-based reward models** (GenRM) (Zhang

et al., 2025) leverage the full language modeling capacity to produce free-form explanations or preference judgments, enabling more flexible feedback but often lacking grounding in external context and struggling with consistency in subjective domains like character dialogue. To understand the limitations in role-playing contexts, we conduct systematic analysis of representative approaches from both paradigms: CharacterRM (Tu et al., 2024) as a scalar-based method and instruction-following models as generative-based approaches (detailed in Appendix C). Our analysis reveals two critical biases that compromise supervision reliability in role-playing tasks (Figure 2).

- **Role Bias.** Evaluation reliability is highly sensitive to character familiarity. As shown in Figure 2(a), main characters achieve both higher human consistency scores (e.g., 0.87 vs. 0.61 under CharacterRM) and markedly lower stability variance (2.1 vs. 14.0) compared to minor characters. This indicates that reward models benefit from abundant pretraining priors when judging well-known or prototypical roles, producing more confident and human-aligned evaluations. In contrast, lesser-known characters lack such priors, leading to unstable assessments where identical responses may receive conflicting scores, with evaluators often misattributing intentions or emotional tone. This instability undermines the reliability of reward signals and systematically disadvantages characters outside the model's training distribution.

- **Reference Bias.** Evaluation outcomes are disproportionately shaped by the presence of explicit reference material. As shown in Figure 2(b), providing character-specific grounding (e.g., persona descriptions, narrative summaries, role guidelines) yields consistently higher human consistency scores (0.79 vs. 0.70 under CharacterRM; 0.59 vs. 0.40 under Reasoning) and significantly lower stability variance (4.1–5.7 vs. up to 16.9). With such scaffolding, evaluators anchor judgments in concrete context, improving alignment with human expectations. Without reference, however, reward models frequently hallucinate motivations, overlook subtle traits, or fall back on generic heuristics, leading to inconsistency and drift. This dependency highlights a critical limitation: conventional reward paradigms fail to robustly internalize character identity, rendering supervision brittle in open-ended or reference-scarce scenarios.

These observations reveal that existing reward models evaluate responses in isolation, without contextual reasoning over character intent or narrative background. Consequently, they fail to capture the multi-dimensional criteria essential for role-playing dialogue, including persona fidelity, emotional appropriateness, and narrative coherence.

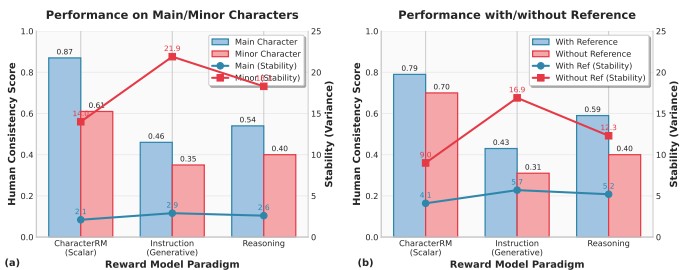

Figure 2: Biases in conventional reward models.

**Reward Modeling as Structured Reasoning.** To address these limitations, we reformulate reward modeling as a reasoning task that integrates multi-step deliberation with character-specific knowledge retrieval. Given a dialogue prompt and two candidate responses, our reward model must not only determine preference but also generate a structured reasoning chain that systematically evaluates responses across multiple quality dimensions. The model accesses retrieved character-specific information from our character-specific knowledge base, enabling deliberative, context-aware comparison analogous to human annotators reasoning over character intent and emotional appropriateness. Moreover, rather than relying on holistic impressions, we guide the model to consider specific aspects of role-playing dialogue quality, including **conversational competence** *(fluency, coherence, consistency)*, **character alignment** *(knowledge exposure, knowledge accuracy, hallucination, persona fidelity)*, and **expressive quality** *(emotional authenticity, engagement, stylistic diversity)*. This systematic consideration ensures comprehensive evaluation while maintaining the flexibility of reasoning-based approaches. The model conducts multi-turn reasoning, retrieving relevant character knowledge as needed, and synthesizes evidence across dimensions to reach well-grounded preference decisions.

**Training Objective and Reward Function.** Inspired by recent advances in reasoning model training (DeepSeek-AI et al., 2025), we apply reinforcement learning to directly optimize the reward model for high-quality comparative analysis, without requiring supervised reasoning traces. We guide learning through a rule-based reward function that integrates three complementary components:

- **Format Reward**. Ensures adherence to structured reasoning format, validating proper use of reasoning tags, retrieval operations, and conclusive preference statements.
- **Answer Reward**. Measures the correctness of predicted preference against ground-truth label using accuracy.
- **Consistency Reward**. Encourages alignment between reasoning and final decisions. Without such supervision, models may produce conclusions inconsistent with their reasoning chains. We train an auxiliary verifier to evaluate reasoning-decision alignment, ensuring that preference choices are well-supported by generated analysis.

While consistency is valuable, integrating it as an independent additive reward introduces a new risk: responses with incorrect answers but strong internal consistency may still receive high total reward. To prevent over-emphasis on consistency, we apply a gated multiplicative formulation where consistency only enhances reward for correct answers:

$$r_{\mathrm{rm}} = r_{\mathrm{ans}} + \lambda_{\mathrm{fmt}_1} r_{\mathrm{fmt}} + \lambda_{\mathrm{cons}}(r_{\mathrm{ans}} \cdot r_{\mathrm{cons}}) - \mu(1 - r_{\mathrm{ans}}) \tag{1}$$

where $r_{\mathrm{ans}} \in \{0,1\}$ denotes answer accuracy, $r_{\mathrm{fmt}} \in \{0,1\}$ format compliance (single boxed conclusion with required tags), and $r_{\mathrm{cons}} \in [0,1]$ the consistency score from an auxiliary verifier. We set $\lambda_{\mathrm{fmt}_1} = \lambda_{\mathrm{cons}} = 0.1$, and apply a mild penalty $\mu = 0.05$ when the answer is incorrect..

**Training with GRPO.** We adopt the Guided Reward Policy Optimization (GRPO) framework (Shao et al., 2024) to train the reward model, combining hard preference labels with soft supervision from auxiliary reward components. Given an old policy $\pi_{\theta_{\mathrm{old}}}$ and a reference policy $\pi_{\theta_{\mathrm{ref}}}$, GRPO optimizes policy $\pi_\theta$ using groups of $G$ rollouts $\{y_i\}_{i=1}^G \sim \pi_{\theta_{\mathrm{old}}}(\cdot|x)$ for each input $x \sim \mathcal{D}$ by maximizing:

$$\mathcal{J}(\theta) = \frac{1}{G} \sum_{i=1}^G \left[ \min\left(\rho_i A_i, \ \mathrm{clip}(\rho_i, \ 1-\epsilon, \ 1+\epsilon) A_i\right) - \beta \, \mathbb{D}_{\mathrm{KL}}(\pi_\theta \parallel \pi_{\theta_{\mathrm{ref}}}) \right] \tag{2}$$

where $\rho_i = \frac{\pi_\theta(y_i|x)}{\pi_{\theta_{\mathrm{old}}}(y_i|x)}$ and $A_i = \frac{r_i - \mu_r}{\sigma_r}$ represents normalized advantages, with $\mu_r$, $\sigma_r$ the mean and standard deviation of $\{r_j\}_{j=1}^G$. The KL penalty $\mathbb{D}_{\mathrm{KL}}$ controls deviation from the reference model. To support retrieval-augmented reasoning, we extend the rollout format to include explicit search operations. During generation, the model issues search queries between `<search>` and `</search>` tags, with retrieved content inserted between `<result>` and `</result>` tags, enabling dynamic knowledge access throughout the reasoning process. Since retrieval results are externally generated, we mask them during gradient computation, ensuring unbiased credit assignment and preventing overfitting to retrieved content.

**Training Data Construction.** We construct a comprehensive reward training dataset that supports reasoning-based evaluation across diverse role-playing scenarios. Our dataset combines multiple sources to ensure broad coverage of character types, dialogue contexts, and evaluation challenges:

- Model-generated data (10K): We sample prompts from ChatHaruhi (Li et al., 2023) and collect responses from 12 diverse dialogue models, including Qwen2.5 series, LLaMA3 series, GPT-4o, and CharacterGLM-6B (Zhou et al., 2024a). For each prompt, we randomly pair response and determine preference through committee voting using GPT-4o, Qwen2.5-72B-Instruct, and CharacterRM. Response pairs with low inter-model agreement are submitted for human annotation to ensure reliability.
- Filtered public data (4K): We augment the dataset with carefully filtered samples from CharacterEval (Tu et al., 2024), retaining only examples with well-defined prompts and reliable preference annotations matching our reasoning-based evaluation framework.
- Human-annotated data (2K). We manually annotate challenging response pairs exhibiting persona misalignment, knowledge hallucination, and emotional inconsistency, with 1K samples for training and 1K reserved for evaluation. These examples particularly benefit from reasoning-based analysis, as they require nuanced understanding of character context and narrative coherence.

To mitigate role bias, we ensure balanced character representation across the dataset, covering both well-known and lesser-known characters from diverse narrative sources. This yields a comprehensive corpus of 15K preference-labeled instances that challenges the reward model to develop robust

reasoning capabilities across varied role-playing contexts. On the held-out evaluation set (1K), our trained reward model achieves **87%** agreement with human preferences, demonstrating reliable capture of nuanced judgments through structured reasoning process.

## 2.3 ROLE-PLAYING AGENT

A central challenge in developing role-playing dialogue agents lies in acquiring high-quality supervision that captures the nuanced requirements of character portrayal. Traditional approaches rely on manually curated dialogue datasets reflecting consistent character behavior, but constructing such supervised data is labor-intensive and difficult to scale across diverse literary characters with complex narrative contexts.

**Reinforcement Learning Framework.** We adopt a reinforcement learning paradigm that circumvents the need for supervised dialogue pairs by leveraging our reasoning-augmented reward model. Instead of learning from fixed examples, the agent receives preference-based feedback reflecting character alignment, narrative coherence, and emotional authenticity. This approach requires only character profiles and user queries, allowing the model to explore response strategies while developing the same reasoning and retrieval capabilities embedded in the reward model.

**Reward and Training.** We adapt the trained pairwise reward model to GRPO training through a scoring transformation that preserves reasoning-based evaluation quality. For each prompt, we sample candidate responses $y_1, ..., y_G$ and compute pairwise comparisons using the reward model's structured reasoning. Each response receives a relative preference score:

$$r_{\text{prefer}_i} = \frac{1}{G-1} \sum_{j \neq i} \mathbb{I} \left[ \text{RM}(x, y_i, y_j) = y_i \right] \tag{3}$$

where $\text{RM}(x, y_i, y_j)$ denotes the reward model's preferred response for prompt $x$ between $y_i$ and $y_j$. These scores are then normalized to obtain advantages used in the GRPO objective. The final reward for response $y_i$ combines preference scores with format compliance ($\lambda_{\text{fmt}_2} = 0.1$):

$$r_{\text{agent}} = r_{\text{prefer}} + \lambda_{\text{fmt}_2} \cdot r_{\text{fmt}} \tag{4}$$

Throughout training, the role-playing agent has access to the same knowledge base as the reward model and is explicitly guided to reason over this context during response generation. The agent learns to formulate contextual queries, retrieve relevant character information, and integrate knowledge through multi-step reasoning processes. This creates a mutually reinforcing dynamic where improved reasoning enhances retrieval effectiveness, and better retrieval enables more sophisticated character understanding. As training progresses, the agent develops increasingly sophisticated reasoning patterns, conducting multi-hop reasoning across character traits, emotional states, and narrative contexts. The model learns to engage in self-reflective reasoning, adjusting response strategies based on retrieved information in ways that mirror authentic character portrayal processes.

**Training Data Construction.** We construct a diverse prompt collection spanning multiple character types and interaction scenarios. Our dataset includes 10K high-quality queries from five novels covering 20 distinct characters in ChatHaruhi and CharacterEval, with balanced representation across character archetypes. Additionally, we manually construct 1K focused prompts targeting three *Dragon Raja* characters to capture nuanced emotional dynamics and complex narrative situations requiring deep character understanding.

## 3 EXPERIMENTS

**Datasets & Evaluation Metrics.** We conduct end-to-end evaluation of the final role-playing agent using CharacterEval (Tu et al., 2024) across 12 metrics in three categories: (1) **Conversational Ability** evaluates general dialogue competence through three core metrics: *Fluency (Flu.)* measures grammatical correctness and natural language flow; *Coherency (Coh.)* assesses topical relevance and logical connection to user prompts; and *Consistency (Cons.)* evaluates internal logical coherence across dialogue turns. (2) **Character Consistency** measures alignment with character identity across

two complementary levels. Knowledge-based metrics include *Knowledge Exposure (KE)*, assessing appropriate demonstration of character-relevant information; *Knowledge Accuracy (KA)*, measuring factual alignment with the character backgrounds; and *Knowledge Hallucination (KH)*, evaluating the model's ability to avoid fabricating contradictory information. The persona-based metrics comprise *Persona Behavior (PB)*, capturing alignment in actions, decision-making patterns, and behavioral consistency; and *Persona Utterance (PU)*, assessing stylistic consistency in speech patterns and linguistic mannerisms. (3) **Role-Playing Attractiveness** reflects the expressive and emotional qualities essential for engaging character interactions, including *Human-Likeness (HL)*, measuring natural and believable character portrayal; *Communication Skills (CS)*, assessing interactive engagement and social competence; *Expression Diversity (ED)*, evaluating varied and dynamic communication styles; and *Empathy (Emp.)*, measuring appropriate emotional responses and empathetic engagement with users. To capture nuanced quality differences, we scale all ratings from the original 5-point scale to a 100-point scale, enabling more precise performance differentiation across models.

**Base LLMs & Baseline Methods.** We compare R4 with three groups of baselines to ensure thorough evaluation: (1) **Instruction models**: representing general-purpose conversational AI, including `GPT-4 Turbo`, `Llama-3.1/3.3 series`, and `Qwen2.5 series`. (2) **Reasoning models**: optimized for multi-step reasoning and complex problem-solving, including `OpenAI o1-mini`, `QwQ-32B-Preview`, `DeepSeek-R1`, and `DeepSeek-R1-Distill variants` across different model scales (Llama-8B/70B, Qwen-7B/32B). These models test whether general reasoning capabilities translate effectively to role-playing scenarios. (3) **Specialized role-playing models**: trained specifically on character-focused dialogue data, including `CharacterGLM-6B`, `Xingchen`[1], `MiniMax`, and `BC-NPC-Turbo` (Tu et al., 2024). These models represent current best practices in dedicated role-playing system development. These baselines establish performance expectations for general-purpose LLMs in role-playing contexts. To ensure fair comparison, all baseline models are built on the same RAG infrastructure, retrieving from an identical curated, character-specific knowledge base with consistent prompts, retrieval policies, and top-$k$ settings.

**Implementation Details.** For the knowledge generation pipeline, we use `gpt-4o-2024-05-13` to perform all related steps, including narrative segmentation based on plot structure and the generation of character-specific knowledge such as persona attributes, internal states, and inferred goals. The reward model is built upon Qwen2.5-32B-Instruct and trained for 2 epochs on our constructed dataset. Similarly, the role-playing agents is based on Qwen2.5-7B/32B-Instruct and also trained for 2 epochs. We initialize from instruction-tuned models rather than base models to ensure more stable reinforcement learning and better final performance—consistent with findings reported in prior work such as ReSearch. The reinforcement learning framework is implemented using Verl Sheng et al. (2025) and ReSearch Chen et al. (2025a). Both the reward model and the role-playing agent share the same retrieval backend. We use multilingual-e5-large Wang et al. (2024a) as the retriever, with indexing and embedding handled by FlashRAG Jin et al. (2024a). Both models query the top-3 retrieved documents per prompt during training and inference. For baseline models comparisons, we directly adopt the implementations and services provided by FlashRAG. All training experiments are conducted on 64 NVIDIA H100 GPUs. Additional training configurations and hyperparameters are provided in Appendix D.

## 3.1 Main Results

Table 1 presents comprehensive performance comparisons between R4 and all baseline models.

**Effectiveness of R4.** On character consistency metrics, R4-32B-Instruct achieves the highest overall performance (64.64 average), significantly outperforming the best baseline (BC-NPC-Turbo at 55.28). Notable improvements include Knowledge Accuracy (+9.52 over the best baseline), Knowledge Hallucination (+2.34), and particularly striking gains in Persona Behavior (+9.8), demonstrating the framework's superior ability to align responses with character-specific behavioral patterns. These substantial improvements validate the effectiveness of reasoning-augmented reward modeling in capturing both factual accuracy and nuanced character portrayal requirements. In conversational ability, R4 models achieve competitive performance with the strongest baselines while maintaining high scores across all dimensions. R4-32B-Instruct matches or exceeds top performers in Coherency

---
[1]https://xingchen.aliyun.com/xingchen

Table 1: Detailed evaluation results. The best performances are highlighted in **bold**.

| Model | Conversational Ability | | | | Character Consistency | | | | | | Role-playing Attractiveness | | | | |
|---|---|---|---|---|---|---|---|---|---|---|---|---|---|---|---|
| | Flu.↑ | Coh.↑ | Cons.↑ | Avg.↑ | KE↑ | KA↑ | KH↑ | PB↑ | PU↑ | Avg.↑ | HL↑ | CS↑ | ED↑ | Emp.↑ | Avg.↑ |
| **Instruct Model** | | | | | | | | | | | | | | | |
| GPT-4 Turbo | 60.60 | 65.80 | 58.00 | 61.47 | 38.20 | 58.40 | 51.00 | 27.80 | 49.60 | 45.00 | 55.60 | 53.60 | 27.20 | 58.80 | 48.80 |
| Llama-3.1-8B-Instruct | 65.60 | 71.00 | 66.20 | 67.60 | 42.40 | 58.80 | 54.60 | 52.20 | 55.40 | 52.68 | 63.60 | 58.80 | 43.60 | 60.40 | 56.60 |
| Llama-3.3-70B-Instruct | 68.00 | 74.00 | 68.60 | 70.20 | 41.20 | 62.40 | 57.40 | 40.80 | 58.40 | 52.04 | 66.20 | 59.80 | 37.00 | 63.20 | 56.55 |
| Qwen2.5-7B-Instruct | 66.80 | 72.00 | 65.40 | 68.07 | 40.60 | 59.60 | 55.60 | 33.40 | 54.80 | 48.80 | 62.20 | 59.60 | 32.00 | 62.80 | 54.15 |
| Qwen2.5-32B-Instruct | 67.80 | 75.20 | 69.20 | 70.73 | 44.20 | 62.00 | 58.80 | 44.00 | 58.00 | 53.40 | 66.00 | 63.00 | 38.60 | 64.20 | 57.95 |
| **Reasoning Model** | | | | | | | | | | | | | | | |
| o1-mini | 68.80 | 72.20 | 57.40 | 66.13 | 38.80 | 49.00 | 56.20 | 25.80 | 50.80 | 44.12 | 64.20 | 48.40 | 23.80 | 59.60 | 49.00 |
| QwQ-32B-Preview | 66.00 | 72.20 | 66.20 | 68.13 | 41.80 | 61.60 | 57.00 | 32.80 | 55.20 | 49.68 | 63.00 | 58.80 | 30.60 | 62.80 | 53.80 |
| Deepseek-R1 | 45.80 | 63.80 | 60.40 | 56.67 | 41.60 | 47.60 | 42.20 | 25.40 | 42.40 | 39.84 | 53.80 | 46.20 | 24.00 | 64.20 | 47.05 |
| DeepSeek-R1-Distill-Llama-8B | 55.40 | 61.60 | 53.60 | 56.87 | 37.60 | 53.00 | 47.00 | 32.60 | 47.20 | 43.48 | 53.60 | 48.00 | 29.80 | 52.20 | 45.90 |
| DeepSeek-R1-Distill-Llama-70B | 62.00 | 69.40 | 62.40 | 64.60 | 39.80 | 59.20 | 53.60 | 28.60 | 52.40 | 46.72 | 59.80 | 55.80 | 28.00 | 59.40 | 50.75 |
| DeepSeek-R1-Distill-Qwen-7B | 60.80 | 63.00 | 63.80 | 62.53 | 38.50 | 54.60 | 51.40 | 20.00 | 53.60 | 43.62 | 49.00 | 54.00 | 25.20 | 63.00 | 47.80 |
| DeepSeek-R1-Distill-Qwen-32B | 64.60 | 71.00 | 65.00 | 66.87 | 40.20 | 61.20 | 55.60 | 26.20 | 54.20 | 47.48 | 61.80 | 56.60 | 26.60 | 61.60 | 51.65 |
| **Specialized model** | | | | | | | | | | | | | | | |
| CharacterGLM-6B | 68.28 | 74.34 | 74.74 | 72.45 | 32.80 | 56.38 | 54.76 | 46.02 | 59.38 | 49.87 | 74.76 | 45.30 | 39.32 | 56.24 | 53.91 |
| Xingchen | 67.56 | 76.14 | 75.08 | 72.93 | 32.72 | 55.36 | 54.86 | 55.44 | 61.10 | 51.90 | 75.14 | 55.44 | 42.00 | 55.98 | 54.64 |
| MiniMax | 72.18 | 78.64 | 76.22 | 75.68 | 36.70 | 58.20 | 58.88 | 55.48 | 62.50 | 54.35 | 75.36 | 53.44 | 43.00 | 60.34 | 58.04 |
| BC-NPC-Turbo | 71.56 | 77.96 | 78.32 | 75.95 | 36.04 | 59.28 | 59.86 | 58.20 | 63.02 | 55.28 | 76.72 | 52.86 | 46.72 | 59.42 | 58.93 |
| **Ours** | | | | | | | | | | | | | | | |
| R4-7B-Instruct | 70.20 | 78.00 | 79.08 | 75.76 | 43.20 | 62.40 | 59.60 | 50.50 | 63.00 | 55.74 | **78.00** | 55.20 | 44.80 | 65.80 | 60.95 |
| R4-32B-Instruct | **74.10** | **82.30** | **80.30** | **78.90** | **48.00** | **68.80** | **62.20** | **68.00** | **76.20** | **64.64** | 77.00 | **63.10** | **50.00** | **69.60** | **64.93** |

(82.30 vs. 78.64 for MiniMax) and Consistency (80.30 vs. 78.32 for BC-NPC-Turbo), while maintaining high Fluency scores (74.10). This demonstrates that reasoning-augmented training enhances character-specific capabilities without compromising fundamental conversational competence. Most remarkably, R4 shows exceptional performance in role-playing attractiveness, where R4-32B-Instruct achieves the highest overall score (64.93 vs. 58.93 for the best baseline). Particularly notable are improvements in Human-Likeness (+0.28), Communication Skills (+0.10), Expression Diversity (+3.28), and Empathy (+5.40). These gains indicate R4's superior capacity for generating emotionally rich, socially engaging responses that create more immersive role-playing experiences.

**Scale Sensitivity of R4.** The cross-model comparison reveals fundamental limitations in existing approaches while confirming R4's framework advantages. Specialized role-playing models (BC-NPC-Turbo) achieve strong attractiveness scores but exhibit weaker character consistency, indicating supervised dialogue training alone cannot capture complex character reasoning requirements. Reasoning models show moderate knowledge-based performance but consistently underperform in persona alignment (avg. 49.68 vs. R4's 64.64), confirming general reasoning capabilities do not transfer effectively to character portrayal. Instruction models demonstrate the most balanced performance among baselines but still fall short across all categories. Notably, model scale amplifies these architectural differences: while baseline performance improvements from scale are modest (e.g., Qwen2.5-7B to 32B: 48.80 vs. 53.40 in character consistency), R4 shows substantial scale benefits (55.74 to 64.64), indicating that the reasoning-retrieval framework becomes increasingly effective with larger parameter counts. This scale sensitivity suggests that R4's unified approach requires sufficient model capacity to fully realize its integration benefits, distinguishing it from simpler supervised or scalar reward approaches.

**Human Evaluation.** Our human evaluation with three annotators ranking 500 dialogue instances confirms that automatic metrics capture meaningful quality differences (Table 2). R4 achieves first rank in 68.2% of cases compared to GPT-4o's 21.6% and CharacterGLM-6B's

Table 2: Human evaluation results.

| Model | Rank 1↑ | Rank 2 | Rank 3↓ | Mean Rank↑ |
|---|---|---|---|---|
| **R4-32B-Instruct** | **68.2 %** | 22.7 % | 9.1 % | **1.42** |
| GPT-4o-2024-05-13 | 21.6 % | **48.4 %** | 30.0 % | 1.94 |
| CharacterGLM-6B | 10.2 % | 28.9 % | **60.9 %** | 2.64 |

10.2%, with statistical significance ($p < 0.01$). Crucially, annotator feedback indicates that R4's advantages stem primarily from superior persona fidelity and narrative coherence—precisely the dimensions our reasoning-augmented approach was designed to address. This alignment between automatic metrics and human judgment validates both our evaluation framework and the practical relevance of our improvements.

**Training Dynamics.** Figure 3 reveals R4's learning process. The consistent reward improvement across training and validation sets demonstrates stable learning dynamics and effective alignment with multi-dimensional character portrayal objectives. The steady increase in response length without

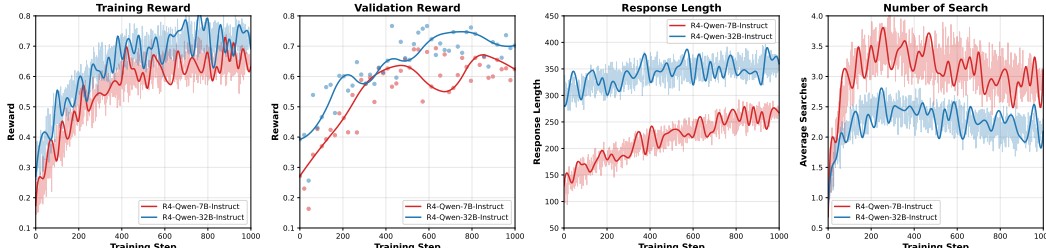

Figure 3: Training dynamics of R4, including training/validation reward, response length, and search count over time.

corresponding quality degradation suggests that R4 develops genuine expressiveness rather than mere verbosity. Similarly, the evolution of retrieval operations (initial increase followed by efficiency gains) indicates that the model learns to formulate more effective queries over time rather than simply retrieving more information.

## 3.2 ABLATION STUDY

To validate each component's contribution, we conduct ablation studies examining both architectural choices and reward modeling approaches.

**Q1: Which Matters More: Reward Quality or Agent Capability?** Table 3 highlights a fundamental asymmetry in the contributions of reasoning and retrieval to role-playing dialogue quality. Our findings show that ***reward model capabilities establish the foundation, while agent capabilities set the upper bound***. Removing reasoning from the reward model (R4-RM w/o Reasoning) leads to a collapse in performance (49.87), nearly identical to the catastrophic failure of removing reasoning entirely (48.23). In contrast, removing reasoning only from the agent (R4-Agent w/o Reasoning) retains substantially higher performance (56.94),

Table 3: Ablation Study.

| Model Variant | Conv. Ability | Character Consistency | Role-Playing Attractiveness | RM Accuracy |
|---|---|---|---|---|
| **Framework Components** | | | | |
| R4 (Full) | **78.90** | **64.64** | **64.93** | **87.00** |
| R4-all w/o Reasoning | 72.15 | 48.23 | 52.84 | 74.20 |
| R4-all w/o Retrieval | 74.82 | 51.67 | 55.19 | 76.80 |
| R4-all w/ GenericRetrieval | 71.94 | 47.95 | 52.33 | 75.10 |
| R4-RM w/o Retrieval | 75.23 | 58.31 | 59.84 | 82.40 |
| R4-RM w/o Reasoning | 70.85 | 49.87 | 53.76 | 76.30 |
| R4-Agent w/o Retrieval | 73.67 | 55.47 | 57.92 | 87.00 |
| R4-Agent w/o Reasoning | 76.12 | 56.94 | 60.18 | 87.00 |
| **Reward Function Components** | | | | |
| R4-RM w/o Consistency | 77.45 | 59.83 | 61.27 | 81.30 |
| R4-RM w/o Format | 78.12 | 62.41 | 63.58 | 84.60 |
| **Alternative Reward Model** | | | | |
| ScalarRM+Agent | 68.34 | 43.89 | 48.72 | 69.80 |
| GenRM+Agent | 69.87 | 45.34 | 50.16 | 71.40 |
| CharacterRM+Agent | 70.52 | 46.87 | 51.93 | 72.90 |

despite operating under the same reward supervision (87% RM accuracy).

This asymmetry yields a key insight for role-playing system design: *supervision quality fundamentally constrains what agents can achieve, regardless of their architectural sophistication*. Even under perfect reward signals, agents without reasoning plateau at 56.94; conversely, agents with reasoning but weaker supervision (76.3% RM accuracy) reach only 49.87. These results suggest a multiplicative rather than additive interaction between reward quality and agent capability.

**Equally important, our analysis shows that simply replacing the reward model in existing systems is not sufficient.** For instance, combining CharacterRM with our dialogue agent yields only 46.87—worse than even our ablated R4 variants (e.g., R4-RM w/o Reasoning: 49.87). This indicates that R4's effectiveness comes not from stronger individual components but from their deliberate co-design. In other words, reward and agent must be engineered to complement each other's reasoning and retrieval processes, rather than swapped in isolation.

**Q2: Is Character-Specific Knowledge Necessary?** The contrast between R4-all w/ GenericRetrieval (47.95) and R4-RM w/o Retrieval (58.31) highlights the critical role of knowledge specificity. Counterintuitively, removing retrieval from the reward model outperforms equipping the entire system with generic retrieval—demonstrating that inaccurate or irrelevant knowledge is more damaging than having none at all. This result overturns common assumptions about knowledge augmentation and underscores a key design principle: effective role-playing systems depend on *precisely curated, character-specific information*, rather than broad but unfocused knowledge access.

## 4 CONCLUSION

We present R4, a unified framework that addresses the fundamental challenge of reliable supervision in role-playing dialogue by equipping both reward models and dialogue agents with reasoning and retrieval capabilities. By formulating reward modeling as a structured reasoning task and integrating retrieval into both supervision and generation, our approach enables fine-grained, persona-aligned evaluation and expressive, contextually grounded dialogue generation.

### ACKNOWLEDGMENTS

This research is supported by the National Natural Science Foundation of China (No.62476127), the Natural Science Foundation of Jiangsu Province (No.BK20242039), the Basic Research Program of the Bureau of Science and Technology (ILF24001), the Scientific Research Starting Foundation of Nanjing University of Aeronautics and Astronautics (No.YQR21022), and the High Performance Computing Platform of Nanjing University of Aeronautics and Astronautics.

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

## A    RELATED WORK

### A.1    RETRIEVAL-AUGMENTED GENERATION

Retrieval-Augmented Generation (RAG) enhances the capabilities of language models by incorporating external knowledge through retrieval mechanisms, grounding responses in factual, domain-specific knowledge (Lewis et al., 2020; Zhao et al., 2024; Wang & Li, 2024a). This approach has been widely studied across several dimensions, including determining when retrieval is necessary (Tan et al., 2024), refining query representations (Mialon et al., 2024; Wang et al., 2023a), compressing and filtering retrieved content (Jin et al., 2024b; Xu et al., 2024a), mitigating noise from retrieved documents (Liu et al., 2024b; Dong et al., 2025b), and improving retrieval quality through instruction tuning (Dong et al., 2025a; Zhou et al., 2024b). Beyond single-step retrieval, more sophisticated multi-step or structured RAG pipelines have been proposed to support complex tasks such as multi-hop reasoning, knowledge planning, and decision-making in domain-specific contexts (Li et al., 2024; Wang et al., 2025; Xi et al., 2025). These methods often integrate search with structured task decomposition to guide model reasoning over intermediate knowledge states. Recent work also explores tighter coupling between retrieval and reasoning through prompt engineering or agentic frameworks. For instance, Search-o1 (Li et al., 2025a) introduces a modular agent that dynamically interleaves retrieval and document-level reasoning. Others employ reinforcement learning to jointly optimize search policies (Zheng et al., 2025; Li et al., 2025b) and reasoning procedures from scratch (Jin et al., 2025; Song et al., 2025; Chen et al., 2025a).

Despite these advancements, current RAG methods remain underexplored in role-playing dialogue settings. Most prior efforts focus on fact-based or reasoning-centric benchmarks, offering limited support for persona-grounded interaction. In particular, they struggle to retrieve and apply character-specific knowledge in emotionally rich or stylistically nuanced conversations—leading to responses that lack consistency with the speaker's persona or fail to sustain user engagement. Our work addresses this gap by leveraging retrieval not just for factual grounding, but as a mechanism to dynamically support character fidelity and dialogue appeal in role-play scenarios.

### A.2    REWARD MODEL FOR REINFORCEMENT LEARNING

Reward modeling (RM) plays a central role in aligning language models with human preferences in reinforcement learning. Early reward models primarily focused on outcome-level supervision—ranking complete outputs based on human preferences (Zhong et al., 2025). However, such models often fail to capture nuanced qualities such as reasoning faithfulness or process transparency. To address these limitations, recent work has explored process reward models (PRMs) that judge the correctness of intermediate steps in chain-of-thought reasoning (Lightman et al., 2024; Setlur et al., 2025; Cui et al., 2025). While effective, PRMs rely on manually annotated intermediate labels or task-specific schemas, limiting their generalizability. To reduce dependence on hand-crafted annotations, generative reward models (GRMs) have been proposed. Models such as Generative Verifiers (Zhang et al., 2025), DeepSeek-GRM (Liu et al., 2025), and JudgeLM (Chen et al., 2025b) frame reward modeling as a next-token prediction task, using generated reasoning chains and voting strategies to produce more interpretable and consistent judgments. Other approaches include prompt-based scoring using instruction-tuned language models (Xiong et al., 2024), and scalar scoring with learned reward heads (Liu et al., 2024a; Lou et al., 2024). While efficient, these methods tend to lack interpretability or rely heavily on prompt design.

Despite these advances, reward modeling remains underexplored in role-playing dialogue, where preferences are shaped not only by correctness, but also by persona alignment, emotional tone, and narrative coherence. Moreover, most reward models operate without access to character-specific knowledge or dynamic reasoning capabilities. In this work, we propose a reward model that integrates reasoning, retrieval, and verifiable feedback. By treating reward modeling as a reasoning task and grounding it in retrieved contextual knowledge, our model provides more accurate and interpretable supervision for training character-aware dialogue agents.

### A.3    ROLE-PLAYING DIALOGUE AGENT

Role-playing dialogue agents aim to generate responses that are not only coherent but also consistent with the persona, background, and emotional state of a given character. Most current approaches

simulate character behavior through prompt engineering or instruction-tuning with character profiles, where models are conditioned on static persona descriptions (e.g., background, tone, or occupation) to emulate specific roles during interaction (Chen et al., 2023; Tu et al., 2024; Zhou et al., 2024a). Further improvements have been made through character-specific fine-tuning, memory-based retrieval, and consistency-oriented generation, which enhance short-term persona fidelity by aligning generated responses with predefined style or factual constraints (Li et al., 2023; Shao et al., 2023; Wang et al., 2024b). However, these methods often focus on surface-level traits (such as tone and wording) and fall short in modeling the underlying cognitive process behind a character's decisions or expressions. Some recent works have explored deeper aspects of role-play, such as simulating internal motivations (Xu et al., 2024b) or reasoning from the character's perspective (Wang et al., 2024c). Yet, most of these rely on human-written references or evaluations, and lack a scalable method for embedding such reasoning capabilities into the generative process itself.

In contrast, our work approaches role-playing as a reasoning task grounded in character-specific knowledge. We go beyond static prompts or stylistic imitation by modeling how agents recall relevant context, reason about their own persona constraints, and produce responses that reflect both situational understanding and consistent character behavior. Our framework explicitly equips the dialogue model with retrieval and reasoning abilities, bridging the gap between surface-level persona simulation and deeper, goal-driven character modeling.

## B  DETAILS OF CHARACTER-SPECIFIC KNOWLEDGE CONSTRUCTION

### B.1  NARRATIVE SEGMENTATION

---

**System Prompt for Narrative Segmentation**

You will receive as input an English or Chinese document with paragraphs identified by 'ID XXXX: <text>'.

**Task:** Find the first paragraph (not the first one) where the content clearly changes compared to the previous paragraphs.

**Output:** Return the ID of the paragraph with the content shift as in the exemplified format: 'Answer: ID XXXX'.

**Additional Considerations:** Avoid very long groups of paragraphs. Aim for a good balance between identifying content shifts and keeping groups manageable.

**Document**:

---

To support retrieval-augmented reasoning in role-playing dialogue, we design a dynamic narrative segmentation procedure tailored for long-form fiction. This module decomposes narrative texts into semantically coherent and contextually self-contained segments, enabling more precise and relevant retrieval during downstream reasoning. Our approach draws inspiration from prior work on LLM-guided segmentation (e.g., LumberChunker (Duarte et al., 2024)), but is adapted to better reflect the episodic, character-centric structure of fictional narratives.

We begin by preprocessing the input novel or story into paragraph-level units, each tagged with a unique identifier. These paragraphs are grouped sequentially into candidate windows $G_i$, such that the total token count of each window remains below a pre-defined threshold $\theta$. The threshold is empirically chosen to balance contextual completeness with model efficiency: large enough to preserve inter-paragraph coherence, yet small enough to avoid context overflow during LLM inference.

For each group $G_i$, we query GPT-4o to determine whether a semantic shift occurs within the window—i.e., whether a paragraph introduces a new event, emotional beat, or dialogue context distinct from the preceding content. This shift point is selected as a segmentation boundary. The next group $G_{i+1}$ starts from the identified boundary, and the process repeats until the entire document is segmented. This iterative, LLM-informed strategy ensures that each resulting chunk reflects a topically unified unit of narrative, improving both retrieval granularity and interpretability in downstream character-centric knowledge synthesis. The prompt used is provided in Table B.1.

### B.2 CHARACTER-SPECIFIC KNOWLEDGE SYNTHESIS

Following narrative segmentation, we construct a structured, character-aligned knowledge repository using the extracted narrative segments. To achieve fine-grained and persona-consistent retrieval and reasoning, we represent each narrative segment as a dynamic, structured "**story-event tree**", inspired by recent advances in hierarchical modeling and dataset construction Wang & Li (2024b;c). Each event tree explicitly captures key narrative dimensions, including temporal structure (*time*), environmental and contextual setting (*scene*), character states and relationships (*character*), and event progression (*event*), thus providing a comprehensive, interconnected representation of the narrative content. The synthesis process employs a systematic four-agent pipeline, where each agent addresses specific aspects of character-centric knowledge construction:

**Knowledge Extraction Agent.** This agent leverages GPT-4o to extract essential factual information from each narrative segment through structured prompting. The extraction process targets five core categories: (1) explicit temporal markers (start and end timestamps, duration, sequence relationships), (2) spatial-environmental contexts (geographical locations, physical settings, atmospheric conditions), (3) causal event chains (action-consequence relationships, trigger events, outcome states), (4) character presence and participation levels, and (5) narrative significance scores. To ensure consistency, we employ a standardized prompt template that constrains the output format to JSON schema with predefined fields, reducing extraction variability across segments.

**Perspective Transformation Agent.** This specialized agent reconstructs narrative events from individual character viewpoints through perspective-aware prompting strategies. For each character present in a segment, the agent generates character-specific interpretations by conditioning on three key factors: (1) the character's established personality profile (accumulated from prior segments), (2) their emotional state trajectory, and (3) their relationship dynamics with other characters. The transformation process systematically infers internal attributes across multiple dimensions: personality traits (introversion/extraversion, risk tolerance, emotional stability), emotional states (anxiety levels, mood valence, arousal intensity), and cognitive patterns (decision-making heuristics, attention focus, memory priorities).

**Mind Agent.** Building upon persona-aligned event representations, this agent synthesizes deeper psychological reasoning through theory-of-mind modeling. The agent explicitly reconstructs each character's internal cognitive processes using structured psychological frameworks: motivation hierarchy (following Maslow's framework), cognitive biases (confirmation bias, attribution patterns), and emotional regulation strategies (coping mechanisms, defense patterns). We prompt GPT-4o with targeted psychological queries such as "Given this character's established personality and current emotional state, what unconscious motivations drive their behavioral choice?" and "What cognitive dissonance or internal conflicts emerge from this situation?" The agent outputs detailed psychological annotations structured as belief-desire-intention (BDI) triplets, enabling downstream reasoning about character behavior patterns.

**Dialogue Extract Agent.** This agent enhances the knowledge repository by identifying and preserving high-fidelity conversational exemplars from the source narrative. Rather than generating synthetic dialogues, the agent employs a two-stage selection process: (1) dialogue identification using linguistic markers (quotation detection, speaker attribution, conversational turn boundaries), and (2) relevance scoring based on psychological significance. The relevance scoring algorithm evaluates dialogues across four criteria: emotional intensity (measured through sentiment analysis and emotional lexicon matching), character revelation (degree of personality or motivation disclosure), relationship dynamics (power shifts, intimacy changes), and narrative pivotality (impact on subsequent

plot development). Selected dialogues are annotated with contextual metadata including speaker emotional states, conversational goals, and implicit subtext interpretations. This approach ensures authentic character voice preservation while providing rich conversational anchors for downstream generation tasks.

**Quality Assurance and Validation** To ensure knowledge quality, we implement a multi-stage validation process: (1) inter-agent consistency checking to identify contradictions across different perspectives, (2) temporal coherence validation to ensure character development follows logical progressions, and (3) psychological plausibility assessment using established personality psychology frameworks. Inconsistencies trigger automated revision cycles where conflicting interpretations are resolved through evidence-based arbitration. This systematic approach ensures knowledge base maintains both factual accuracy and psychological authenticity.

The output from all these agents—structured facts, character-perspective interpretations, psychological annotations, and dialogue exemplars—are integrated into a unified character-specific knowledge base, organized into a composable and indexed event-tree structure. This comprehensive knowledge representation enables precise multi-dimensional retrieval nuanced reasoning during downstream dialogue generation and reward assessment tasks, significantly improving the realism and the consistency of the generated role-playing interactions.

## C  DETAILS OF BIASES IN REWARD MODEL

To systematically investigate biases in reward models used for role-playing dialogue evaluation, we conducted a detailed comparative study between two representative reward modeling paradigms: scalar-based reward modeling and generative-based reward modeling. Specifically, we compared CharacterRM (scalar-based) with instruction-tuned Qwen2.5-7B-Instruct and reasoning-enabled generative models DeepSeek-R1-Distill-Qwen-7B.

We constructed a controlled evaluation corpus based on two representative fictional universes: *Dragon Raja* and *Harry Potter*. We selected two main characters—Lu Mingfei (*Dragon Raja*) and Harry Potter (*Harry Potter*)—and four secondary characters: Hui Liyi, Xia Mi, Luna Lovegood, and Severus Snape. Each character was associated with 100 dialogue-response evaluation pairs, evenly covering diverse interaction contexts and emotional complexities. To establish a reliable reference, the evaluation pairs were manually reviewed by the authors.

## D  IMPLEMENTATION DETAILS

For the knowledge generation pipeline, we use `gpt-4o-2024-05-13` to perform all related steps, including narrative segmentation based on plot structure and the generation of character-specific knowledge such as persona attributes, internal states, and inferred goals. The reward model is built upon Qwen2.5-32B-Instruct and trained for 2 epochs on our constructed dataset. Similarly, the role-playing agents is based on Qwen2.5-7B/32B-Instruct and also trained for 2 epochs. We initialize from instruction-tuned models rather than base models to ensure more stable reinforcement learning and better final performance—consistent with findings reported in prior work such as ReSearch. The reinforcement learning framework is implemented using Verl (Sheng et al., 2025) and ReSearch (Chen et al., 2025a). Both the reward model and the role-playing agent share the same retrieval backend. We use multilingual-e5-large (Wang et al., 2024a) as the retriever, with indexing and embedding handled by FlashRAG (Jin et al., 2024a). Both models query the top-3 retrieved documents per prompt during training and inference. For baseline models comparisons, we directly adopt the implementations and services provided by FlashRAG. All experiments were conducted on 64 NVIDIA H100 GPUs, with full parameter optimization. For reinforcement learning of reward model and dialogue agent, we used a learning rate of $1 \times 10^{-6}$ and a training batch size of 64. The model was trained for 2 epochs, with each prompt generating 5 rollouts during the sampling phase. We set the rollout temperature to 1.0 to balance exploration and response diversity. Additionally, the KL loss coefficient is 0.001 and clipping ratio is 0.2 to constrain policy updates and ensure training stability Wang et al. (2023b).

---

Case study of R4's role-playing agent: Case 1 (Romantic Ambiguity and Persona Adherence)

---

**Character**: Lu Mingfei

**User Question**: Do you love Nono?

---

**Agent Response:**

<think> This is a direct question about Lu Mingfei's romantic feelings—a core emotional arc in the story. Lu Mingfei doesn't openly confess. He often avoids the topic or hides behind humor and self-deprecation. Still, he can't outright deny it—doing so would break character consistency. A proper response needs to balance emotional tension: denial on the surface, vulnerability underneath, wrapped in sarcasm or jokes. </think>

<search> Lu Mingfei's feelings toward Nono </search>
<result> Lu Mingfei secretly likes Nono but never confesses directly... </result>

<think> The answer should reflect that: - He won't say "yes" outright; - He'll joke, then slip in a wistful, subtle admission; - Tone should shift between light sarcasm and buried longing; - End on ambiguity, leaving the user to guess the real feeling. </think>

<answer> Like her? She's the kind of girl who walks in sunlight with her own background music. Me standing next to her? It's like the local ramen guy photobombing a global fashion shoot—do you think I could like someone like that?

Sure, I fantasized—so much I could write a 25-episode drama. No ads included. But in real life... I just sneak glances and pretend I'm looking at the vending machine. So... I don't know. </answer>

---

Case study of R4: Case 2 (Self-Reflection and Character Growth)

---

**Character**: Lu Mingfei

**User Question**: Why did you go on that mission involving the Sheqi Bajia? I thought you were afraid of dying?

---

**Agent Response:**

<think> User's question touches a core contradiction in Lu Mingfei: his fear of death versus his pattern of self-sacrifice. On the surface, he's someone who avoids danger. My first instinct is to say he didn't volunteer—he was pushed into it. </think>

<search> Lu Mingfei motivation Sheqi Bajia mission </search>
<result> Initially hesitant and fearful; attempted avoidance... Later learned of the mission's link to Chen Motong, his brother ... </result>
<search> Lu Mingfei emotional reaction to Chen Motong </search>
<result> Shift from avoidance to personal responsibility... Expressed guilt and emotional paralysis upon learning of Chen's death... Quote: "I'm not a brave man, but sometimes even cowards get pushed forward." </result>

<think> Wait—that contradicts my initial assumption. He did go willingly in the end, out of guilt and unresolved family ties. Saying he only went because he was forced would ignore his emotional growth and deeper motivations. The final answer should reflect this progression: from fear to acceptance, driven by emotional weight. </think>

<answer> I'm afraid of dying—of course I am. You think I didn't consider faking a fever to skip it? But that time was different. I learned it involved my brother, Chen Motong—and he was already gone. I just... froze. Sometimes you do things you're not ready for, because not doing them would be worse. I didn't get brave—I just didn't want to regret it later. </answer>

---

# E    CASE STUDY

We present two cases where the agent role-plays *Lu Mingfei* from *Dragon Raja*.

In the first case (Table D), when asked "Do you love Nono?", the agent avoids direct confession, instead responding with humor, self-deprecation, and emotional ambiguity—faithfully reflecting Lu Mingfei's canonical behavior. This illustrates the model's ability to maintain nuanced character consistency and emotional appeal.

In the second case (Table D), the user challenges the agent's motivations: "Why did you go on that mission? I thought you were afraid of dying?" Initially, the model assumes fear-based reluctance, but after retrieving relevant context (e.g., family ties), it revises its reasoning and generates a more accurate, emotionally grounded response. This self-correction process highlights an emergent *self-reflection* ability, where retrieval informs dynamic reasoning adjustment.

These examples showcase R4's capacity to blend character grounding, retrieval-augmented reasoning, and high-level behavioral coherence—key to effective role-playing dialogue.

