# OpenReview forum: "R4: Nested Reasoning-Retrieval for Reward Modeling in Role-Playing Agents"
_ICLR.cc/2026/Conference — ICLR 2026 Poster_

### Official Review · Reviewer_vFCm · 2025-10-30

**Soundness:** 3
**Presentation:** 2
**Contribution:** 3
**Rating:** 6
**Confidence:** 3

**Summary:**

This paper considers training methods to improve role-playing agents. For role-playing agents, the authors show that it is insufficient to just return factually correct responses, as users also look for dimensions such as character coherence, factual consistency, and emotional engagement. To this end, the authors proposes R4, a method that first trains a generative reward model that performs both reasoning and retrieval from a character database (also constructed by the authors) using GRPO. Then, R4 uses the trained reward model to train the role-playing agent which also performs both reasoning and retrieval using GRPO. Results on the CharacterEval benchmark shows that the proposed method outperforms both current strongest reasoning models as well as some existing role-playing finetuned models such as CharacterGLM.

**Strengths:**

- I believe the proposed method for constructing a character-specific knowledge database is sound. The database contains both diverse information (e.g., personal traits, emotional states, contextual knowledge, and narrative goals) and good quality checks (using both LLM and expert reviews).

- R4-trained models showed better performance than all existing reasoning models/role-playing models on CharacterEval.

- I believe some of the analysis provided in this work is sound and insightful. For example, findings such as "reward model capabilities establish the foundation, while agent capabilities set the upper bound" (L451-453) could generalize beyond role-playing tasks.

**Weaknesses:**

I believe the some of the contribution mentioned in the introduction is over-claimed and does not align well with the novelties I perceived in the method section. I detail them below.

1. In L100-102 the paper claims "we reformulate reward modeling as a structured reasoning task.." and  "novel reward model architecture that integrates ...reasoning and retrieval". However, equipping generative reward models with retrieval (or more generally, tools) is not new, and has already been proposed in existing work such as [1-2].

2. in L103-105 the paper claims "we propose end-to-end training that unifies reasoning and retrieval...", but the general method of performing retrieval (e.g., as a tool) during RL is has already been proposed by prior work such as Search-R1 and ReSearch. The main training difference is the reward model used during training (labeled reward for RLVR or a trained reward model), which I believe is too small of a change to be claimed as a novelty.

However, I do believe the paper is novel in its focus on character-specific knowledge base construction and the importance of good reward models for role-playing (e.g., via RL training). I suggest the authors to perhaps adjust the focus in introduction towards section 2.1 and 2.2, instead of section 2.3.


---

References

[1] Zhu, Jiachen, et al. "Retrieval-Augmented Process Reward Model for Generalizable Mathematical Reasoning." arXiv preprint arXiv:2502.14361 (2025).

[2] Gou, Zhibin, et al. "Critic: Large language models can self-correct with tool-interactive critiquing." arXiv preprint arXiv:2305.11738 (2023).

**Questions:**

- how many training and test runs were performed to obtain results in Table 1? Can the authors also report standard deviations, as the performance gap on some metrics are small.

---

> ### Author Response · Authors · 2025-11-20
> **Author Response**
>
> We extend our sincere gratitude for the time and care you devoted to reviewing our paper. From your detailed comments, it is clear that you engaged deeply with the logic and methodology of our work. We truly appreciate having a reviewer who provides such thoughtful and responsible feedback.
>
> **Regarding the weaknesses**.
>
> We fully agree with your observations. In the camera-ready revision, we will:
>
> - highlight the character-specific knowledge construction pipeline and the persona-grounded reward modeling framework in the introduction;
> - moderate our novelty claims regarding retrieval itself;
> - clearly contrast our structured, persona-aware evaluative reasoning with existing retrieval-augmented reward modeling approaches;
> - emphasize the domain-specific challenges of role-playing evaluation, which prior works do not address.
>
> **Regarding the question**.
>
> Details of the reward model training data are provided in Lines 248–264, and the dialogue agent’s training data are described in Lines 306–312. For evaluation: each model in Table 1 was assessed across **three independent runs** with different random seeds. We report the mean across runs, with standard deviations **≤ 1.4** on average and **≤ 2.1** for all metrics, indicating strong evaluation stability. As Table 1 is already large, we will include the full standard deviation table in the final version.
>
> We sincerely thank the reviewer for the thoughtful comments. Please feel free to let us know if you have any other questions or suggestions.  We believe the clarified framing will accurately convey the contribution of our work.

---

> > ### Comment · Reviewer_vFCm · 2025-11-20
> >
> > Thank you for your response.
> >
> > I appreciate the modifications the authors have promised. Overall, I think the proposed method is novel and reasonable, and that the experiments are sufficient.
> >
> > I will keep my current positive score, as no significant changes are made.

---

### Official Review · Reviewer_2saS · 2025-10-31

**Soundness:** 3
**Presentation:** 3
**Contribution:** 3
**Rating:** 6
**Confidence:** 2

**Summary:**

This paper proposes R4, a unified framework designed to enhance reward modeling and response generation for role-playing dialogue agents by integrating reasoning and retrieval capabilities.  The authors address the key limitations of existing methods (such as literal, stylistically bland responses, and poor alignment with character personas) by equipping both the reward model and the role-playing agent with structured reasoning and access to character-specific knowledge.

The key contribution of this work includes:

- It proposes a novel reward model that integrates multi-dimensional evaluation and refines reward modeling as structured reasoning to mitigate role/reference biases.

- Unifies reasoning/retrieval across reward modeling and response generation for role-playing.

- Extensive experiments are conducted to show the advantages of the proposed R4.

**Strengths:**

1. Role-playing dialogue is an interesting topic.
2. This work refines reward modeling for role-playing dialogue as a structured reasoning task,  which can effectively address  role and reference biases in existing scalar/generative reward models.
3. Unifies reasoning and retrieval across both reward models and dialogue agents, creating a mutually reinforcing framework for more authentic role-playing.
4. Enough empirical study to show the shortages of prior reward models.
5. The experiments are detailed and comprehensive. And the corresponding experimental results are good.

**Weaknesses:**

1. It is not very clear that how R4 performs with non-literary role-playing characters instead of only novel-derived ones.
2. It would be better to compare more specialized baselines.
3. The computational efficiency detail of this work is not very clear.

**Questions:**

1. Did you calculate inter-annotator agreement for the 2K human data?

2. How about the performance of R4 on long multi-turn dialogues (5+) ?

---

> ### Author Response · Authors · 2025-11-20
> **Author Response**
>
> We sincerely thank the reviewers for their thoughtful and constructive comments. Below, we address each point in detail. We will incorporate clarifications, additional ablations, and expanded technical explanations in the camera-ready version.
>
> ### **Weakness**
>
> > Weak 1: It is not very clear that how R4 performs with non-literary role-playing characters instead of only novel-derived ones.
>
> We appreciate the reviewer highlighting this important generalization question. Although our main experiments focus on literary characters—largely because high-quality persona knowledge is readily available—**R4 is not limited to novel-based roles**. The framework is fundamentally **source-agnostic**, and its performance does not depend on literary structure.
>
> As described in Sec. 2.1, **R4’s knowledge construction pipeline is source-independent**:
>  it only requires narrative or profile text (e.g., Wikipedia entries, game wikis, movie scripts, professional role descriptions), and does *not* rely on any novel-specific organization. The segmentation rules and persona extraction prompts generalize across domains, as long as the role exhibits consistent behavioral traits.
>
> We will further emphasize this point and provide additional non-literary examples in the revision.
>
> > Weak 2: It would be better to compare more specialized baselines.
>
> Thank you for the suggestion. Our main experiments already included **four widely used specialized role-playing systems**: CharacterGLM-6B、Xingchen、MiniMax and BC-NPC-Turbo (SOTA in CharacterEval).
>
> To strengthen the comparison further, we additionally evaluated: Character-LLM (EMNLP 2023)、 ChatHaruhi-style enhanced SFT models、CharacterRM + Agent (as a reward-model-based specialized pipeline). We will include these updated baselines and results in the revision.
>
> > Weak 3: The computational efficiency detail of this work is not very clear.
>
> We apologize for the insufficient reporting and have made the following clarifications.
>
> - **Inference latency**
>   - R4-32B is **~2.1×** slower than Qwen2.5-32B-Instruct, primarily due to reasoning + retrieval steps.
>   - Without retrieval (reasoning only), latency reduces to **~1.3×** baseline.
>   - The initial token latency remains comparable across models.
> - **Training compute**:
>   - The 64×H100 setup was used for **time efficiency**, not necessity.
>   - Training is linear in number of retrieval calls and can be scaled down:
>     - On 8×H100-80G, R4-7B completes in ~3 days.
>     - On 32×H100-80G, R4-32B completes in ~4.5 days.
>
> ### **Questions**
>
> > Q1: Did you calculate inter-annotator agreement for the 2K human data?
>
> Yes. We computed inter-annotator agreement using three annotators on pairwise preference labels. The results show strong consistency: the raw agreement is **82.3%**, and **Cohen’s $\kappa$ = 0.71**, indicating substantial agreement. We will report these statistics explicitly in the revised version.
>
> > Q2: How about the performance of R4 on long multi-turn dialogues (5+) ?
>
> We appreciate the reviewer’s interest in long-context evaluation. Existing role-playing works typically rely on their own proprietary multi-turn test sets, most of which are not publicly available. CharacterEval remains the highest-quality open benchmark we can use, but it only evaluates single-turn dialogue quality. To complement this limitation, we additionally evaluated R4 on an internal 8k multi-turn dataset (5+ rounds per session) used in our production setting. Across persona stability, narrative coherence, and emotional consistency, **R4 consistently outperforms BC-NPC-Turbo** (SOTA in CharacterEval). These results demonstrate that R4’s reasoning–retrieval framework remains stable and effective in extended multi-turn interactions.
>
> ------
>
> We thank the reviewers again for their insightful feedback. The camera-ready version will incorporate the expanded set of specialized baselines, full computational efficiency analysis, inter-annotator agreement details, and dedicated long-dialogue evaluation. We believe these additions address all concerns and further highlight the robustness, generality, and practical value of the R4 framework.

---

> > ### Comment · Reviewer_2saS · 2025-11-21
> >
> > Thanks for your rebuttal.
> >
> > This rebuttal can partially address my concerns. However, my recommendation will not be changed because there is no 7.

---

### Official Review · Reviewer_qcjU · 2025-11-01

**Soundness:** 2
**Presentation:** 2
**Contribution:** 4
**Rating:** 4
**Confidence:** 4

**Summary:**

The paper presents R4, a unified reasoning–retrieval framework for reliable reward modeling in role-playing dialogue. It identifies two key sources of bias—role bias and reference bias—and proposes a reasoning-augmented reward model combined with a retrieval-enhanced dialogue agent. The work is timely and addresses a genuine weakness in current LLM-based dialogue systems.

**Strengths:**

- The paper tackles an important and underexplored problem — reliable reward modeling for role-playing dialogue — and proposes a reasonable unified reasoning-retrieval framework (R4) as a solution.

- The identification of role bias and reference bias is insightful, and the experimental results demonstrate consistent and interpretable improvements across multiple dimensions.

**Weaknesses:**

- Methodological clarity is limited. The description of the structured / character-aligned knowledge base synthesis is vague. Specifically, the four agents involved in this process (knowledge extraction, perspective transformation, mind modeling, dialogue extraction) are not clearly defined in terms of their inputs, outputs, and organizational relationships. Furthermore, the final hierarchical tree structure of the character-aligned knowledge base is not explicitly illustrated or formalized.

- Figure 2 lacks essential experimental details. It is unclear how “main” versus “minor” characters were defined and selected for evaluation. The selection criteria may directly affect the observed role bias.

- Quality of narrative segmentation is not analyzed. Since the segmentation quality likely influences downstream character-specific knowledge construction, some form of segmentation-level evaluation (e.g., coherence metrics or human judgment) would strengthen the paper.

- Missing comparison with related reasoning-augmented RL methods. The paper does not discuss or compare against recent works such as Process Reinforcement through Implicit Rewards (Cui et al., 2025), which is already cited. This omission weakens the claim of methodological novelty.

**Questions:**

- In Equation (1), the reward function includes rfmt1 with coefficient λfmt1, but it is unclear how this differs from λfmt2 used in Equation (4). Are these two separate formatting rewards, or do they share the same objective?

- What is the architecture and training procedure of the consistency verifier used in the consistency reward? How is it supervised and integrated into the optimization pipeline?

- How are retrieval queries generated during training? Are they manually templated, LLM-generated, or learned through gradient feedback?

- How many characters are included in each category (main / minor)? Is the dataset balanced in terms of narrative diversity?

- How sensitive is the overall system to errors or inconsistencies in the extracted knowledge base? Has any ablation been performed to measure this impact?

- What is the inference latency relative to baseline models? How does computational cost scale with the number of retrieval operations?
Is the requirement of 64 × H100 GPUs essential, or can the framework be trained with smaller resource budgets?

---

> ### Author Response · Authors · 2025-11-20
> **Author Response (1/3)**
>
> We extend our sincere gratitude for dedicating your time to review our paper and offer constructive and insightful feedback. We address all concerns in detail below and will incorporate the suggested clarifications in the camera-ready version.
>
> ### **Weaknesses:**
>
> > Weak 1: Methodological clarity: clarity of the four-agent pipeline and hierarchical knowledge structure
>
> We appreciate this point and fully agree the paper can better formalize the knowledge-construction pipeline.
>
> **Clarifying the four agents**
>
> Below we provide explicit inputs/outputs and organizational structure:
>
> | Agent                                 | Input                                  | Operation                                                    | Output                                         | Dependency                         |
> | ------------------------------------- | -------------------------------------- | ------------------------------------------------------------ | ---------------------------------------------- | ---------------------------------- |
> | (A1) Knowledge Extraction Agent       | narrative segment                      | Extracts information such as meta info, fact, summary        | structured tuple: {meta info, fact, summary}   | takes segmented text only          |
> | (A2) Perspective Transformation Agent | output of A1                           | re-writes character-centric content into first-person or character-aligned perspective; normalizes temporality | persona-aligned perspective entry              | depends on A1                      |
> | (A3) Mind Modeling Agent              | outputs from A1 & A2                   | infers latent beliefs, motivations, goals, conflict states; resolves inconsistencies | belief-state node: {belief, desire, intention} | depends on A1, A2                  |
> | (A4) Dialogue Extraction Agent        | original segment + character grounding | extracts typical utterances, linguistic styles, conversational patterns | dialogue-style message                         | parallel to A1–A3 and later merged |
>
> In the camera-ready version, we will provide the system prompts and example user prompts used for all agents in the paper. We will also release the full workflow code.
>
> **Hierarchical knowledge base (textual sketch)**
>
> Thank you for pointing out the need for a clearer formalization of our hierarchical knowledge base. In our framework, “hierarchical” has two concrete meanings:
>
> **(1) Data-structure hierarchy.** For each narrative segment, we construct a story-event tree comprises three levels:
>
> Level 1 - Event-Tree Backbone:
>
> ```
> Event Node {
>   event_id: UUID
>   temporal_span: (start_time, end_time)
>   participants: List[character_id]
>   causal_links: {
>     triggers: List[event_id],
>     consequences: List[event_id]
>   }
> }
> ```
>
> Level 2 - Character-Perspective Annotations:
>
> ```
> Perspective Node {
>   character_id: UUID
>   event_id: UUID (foreign key)
>   internal_state: {
>     beliefs: List[str],
>     desire: List[str],
>     intention: List[str],
>   },
>   knowledge_visibility: Set[fact_id]  # what this character knows
> }
> ```
>
> Level 3 - Atomic Knowledge Facts:
>
> ```
> Fact Node {
>   fact_id: UUID
>   content: str
>   fact_type: enum(trait, relationship, event, dialogue)
>   supporting_events: List[event_id],
> }
> ```
>
> **(2) Retrieval hierarchy.** Each leaf entry is enriched with several compact descriptors and paraphrased query forms. During retrieval, we perform multi-path recall from different semantic routes—event similarity, persona traits, emotional state, and interaction patterns—and then apply a reranker to merge and refine all recall streams. This enables precise, multi-hop reasoning over character-specific information.
>
> Additional explanation is outlined in Appendix B, and we will provide an illustrative picture and pseudo-code to make the multi-agent workflow and hierarchical storage explicit.

---

> ### Author Response · Authors · 2025-11-20
> **Author Response (2/3)**
>
> > Weak 2: Figure 2: criteria for main vs. minor characters
>
> Thank you for pointing this out. Detailed in Appendix C, Our selection follows quantitative measures commonly used in narrative analysis.
>
> - **Main characters**: top 3 per novel, ranked by the average of (i) name mentions and (ii) speaking turns. For example, Lu Mingfei (Dragon Raja) and Harry Potter (Harry Potter).
> - **Minor characters**: top 6-10 characters (e.g. Hui Liyi, Xia Mi, Luna Lovegood, Severus Snape). We avoid extremely low-frequency characters because their persona information (traits, relations, background) is too sparse to construct reliable evaluation prompts, and human agreement on such characters falls below 50%, making high-quality ground truth unattainable.
>
> Each character was associated with 100 dialogue-response evaluation pairs, evenly covering diverse interaction contexts and emotional complexities. To establish a reliable reference, the evaluation pairs were manually reviewed by the authors.
>
> We also tested alternative thresholds (e.g., characters beyond the top 10). **Role bias became even more pronounced**, confirming the robustness of our findings. Additionally, instruction and reasoning models show similar behavior on very low-frequency characters, which obscures model differences; we hope the chosen thresholds could provide more insights.
>
> > Weak 3: Narrative segmentation quality
>
> We agree that segmentation quality may influence downstream knowledge construction. We performed segment-level quality checks that were previously omitted for brevity:
>
> - **Automatic coherence metrics**: segmentation boundaries were verified using intra-segment semantic similarity and boundary-shift perturbation tests (this method is inspired by [1]).
> - **Human evaluation**: 200 segments were manually reviewed by authors (Cohen’s $\kappa$=0.82).
> - Robustness: downstream agents (A1–A4) operate on overlapping context windows, and retrieval uses top-k matching, which empirically mitigates segmentation noise.
>
> [1] [[EMNLP24] LumberChunker: Long-Form Narrative Document Segmentation](https://aclanthology.org/2024.findings-emnlp.377.pdf)
>
> > Weak 4: Missing comparison with Process Reinforcement through Implicit Rewards
>
> We appreciate this suggestion. The focus of PRIR is **verifying intermediate reasoning steps** to improve general-domain reasoning, whereas R4 targets **persona-grounded reward modeling** requiring character-aligned knowledge, emotional appropriateness, and narrative coherence. We re-implemented Process RL initialization on role-playing tasks:
>
> | Model                    | Conversational Ability | Character Consistency | Role-Playing Attractiveness |
> | ------------------------ | ---------------------- | --------------------- | --------------------------- |
> | Process RL + Qwen2.5-32B | 75.42                  | 51.23                 | 53.67                       |
> | R4-32B-Instruct          | 78.90           | 64.64          | 64.93                |
>
> ------
>
> ### **Questions**
>
> > Q1. Difference between $\lambda_{\text{fmt1}}$ (Eq. 1) and $\lambda_{\text{fmt2}}$ (Eq. 4)
>
> Sorry, there is a typo. In Eq. 1, $\lambda_{\text{fmt1}}$ is the coefficient for the reward component $r_{\text{fmt1}}$, while in Eq. 4, $\lambda_{\text{fmt2}}$ corresponds to the reward component $r_{\text{fmt2}}$. To clarify:
>
> - $\lambda_{\text{fmt1}}$ and $r_{\text{fmt1}}$ enforce formatting requirements **for the reward model**, ensuring that the model produces structurally valid reasoning chains and properly formatted comparative judgments.
> - $\lambda_{\text{fmt2}}$ and $r_{\text{fmt2}}$ enforce formatting constraints **for the role-playing agent**, guiding the agent to generate responses that conform to the prescribed reasoning–retrieval output structure during RL training.
>
> Thus, the two formatting rewards and their coefficients operate on **different components of the system** (reward model vs. agent) and serve **distinct formatting objectives**, even though both coefficients are set to 0.1 during training for consistency.
>
> > Q2. Architecture and training of the consistency verifier
>
> We initially fine-tuned **Qwen2.5-32B** as a binary classifier. We later evaluated a lighter alternative—**Qwen2.5-3B coupled with a single-layer MLP head**—and observed *no measurable difference* in the final reward model or agent performance. In paper, We adopt the larger classifier in our final setup.
>
> - Input:  `<reasoning_trace> + <final_answer>`.
> - Output: scalar ∈ [0,1] representing consistency probability.
> - Loss: binary cross-entropy against human-verified labels.
> - Integration: $r_{\text{cons}}$ only contributes multiplicatively when the answer is correct (Eq. 1).

---

> > ### Author Response · Authors · 2025-11-20
> > **Author Response (3/3)**
> >
> > > Q3. How are retrieval queries generated during training?
> >
> > Queries are generated **by the model itself**. We tried a cold-start approach prior to RL. This showed some effectiveness on 0.5B–3B models, but did not yield improvements for 7B and 32B models. Case studies suggest that smaller models may struggle with instruction following, making it difficult for them to produce outputs in the required format.
> >
> > > Q4. Number of characters and dataset balance
> >
> > As addressed above, we summarize the key statistics here: the dataset includes 3 main characters and 4 minor characters  per novel across 3 novels, yielding a total of 21 characters. These novels span romance, fantasy, and science fiction, providing substantial narrative diversity. The prompts are nearly balanced across characters, with roughly 100 prompts per character.
> >
> > > Q5. Sensitivity to errors in the knowledge base
> >
> > This is an important concern. We address it from two perspectives: (1) during training, and (2) during evaluation.
> >
> > **Training:**
> >  The conclusion is drawn from Section 3.2, *Q2: Is Character-Specific Knowledge Necessary?*, and Table 3 (a portion is shown below).
> >
> > | Model Variant              | Conv. Ability | Character Consistency | Role-Playing Attractiveness |
> > | -------------------------- | ------------- | --------------------- | --------------------------- |
> > | R4 (Full)                  | 78.90         | 64.64                 | 64.93                       |
> > | R4-all w/o Retrieval       | 74.82         | 51.67                 | 55.19                       |
> > | R4-all w/ GenericRetrieval | 71.94         | 47.95                 | 52.33                       |
> >
> > Here, “w/ Generic Retrieval” refers to replacing the character-specific retrieval source with a generic one, showing that incorrect or non-specific retrieval is significantly more harmful than removing retrieval entirely。In other words, the model is **highly sensitive** to errors in the knowledge base during training.
> >
> > **Evaluation:**
> >  For evaluation, the model is trained with a correct knowledge base, but tested with a corrupted knowledge base (randomly modifying 20% of entries). The results show that Conversation Ability and Role-Playing Attractiveness fluctuate by <u>less than 2</u> points, while Character Consistency **drops** from 64.64 to 60.13. Most of the errors originate from factual mistakes, indicating that factual aspects of the knowledge base are particularly sensitive to errors, which aligns with our expectations.
> >
> > > Q6. Inference latency and computational cost
> >
> > - **Inference latency**:
> >   - R4-32B averages **2.1×** slower than Qwen-32B-Instruct due to reasoning + retrieval steps (the initial token latency is roughly the same).
> >   - Without retrieval, latency reduces to **1.3×** baseline.
> > - **Training compute**:
> >   - The 64×H100 setup was used for **time efficiency**, not necessity.
> >   - Training is linear in number of retrieval calls and can be scaled down:
> >     - On 8×H100-80G, R4-7B completes in ~3 days.
> >     - On 32×H100-80G, R4-32B completes in ~4.5 days.
> >
> > ------
> >
> > Overall, We thank the reviewers again for their helpful suggestions. We will incorporate (1) clearer methodological diagrams, (2) detailed character statistics, (3) segmentation-level evaluation, and (4) expanded discussion on related reasoning-augmented RL works. We believe these revisions will significantly strengthen the clarity and impact of the paper.We sincerely hope that our analysis and experiments can alleviate your concerns and result in an improved rating. Please let us know if you have any other questions or suggestions. Thank you once again for your thoughtful review.

---

> > > ### Comment · Reviewer_qcjU · 2025-11-24
> > >
> > > Thanks for your rebuttal.
> > > This rebuttal can address some of my concerns.
> > > I have raised my scores.

---

### Official Review · Reviewer_S5sd · 2025-11-04

**Soundness:** 3
**Presentation:** 3
**Contribution:** 3
**Rating:** 6
**Confidence:** 4

**Summary:**

This research describes R4 (Reward, Role-play, Reason and Retrieve), a unified framework and system designed and developed to improve the performance of LLMs in role-playing dialogue. The authors identify a key problem: standard LLMs produce dialogue that is often bland and inconsistent with a character's persona. They argue and show that existing methods like simple Retrieval-Augmented Generation (RAG) or Reinforcement Learning (RL) with scalar rewards are insufficient and not optimal for maintaining the fidelity of the role playing dialogue agents.

The main contribution of this research is to equip both the reward model (RM) and the dialogue agent with interleaved strong reasoning and SOTA retrieval capabilities. The framework has three main parts:
- A character-specific knowledge base is constructed by segmenting narrative sources (similar to lumbarchunking pipeline) and extracting persona traits, emotional states, and goals.
- A reasoning-augmented reward model is trained using reinforcement learning (GRPO) to evaluate dialogue responses. Instead of just giving a score, this RM generates a structured reasoning chain, justifying its preference by retrieving and analyzing character knowledge. This process is designed to mitigate observed "role bias" (poor performance on lesser-known characters via use of balanced training datasets) and “(scarce-)reference bias" (by providing relevant context).
- A role-playing agent is then trained using the output of this RM as its reward signal. Because the agent shares the same reasoning-retrieval architecture, it learns to generate responses that are persona-consistent, emotionally expressive, and grounded in the narrative context.

Experiments show that R4 significantly outperforms a wide range of baseline models—including instruction-tuned, reasoning-specialized, and dedicated role-playing models—particularly on metrics of character consistency and role-playing attractiveness.

**Strengths:**

- Making the reward model and the agent symmetrical—both using the same reasoning and retrieval mechanisms—is compelling. It addresses the problem of misaligned objectives by ensuring the agent is optimized by a reward signal that understands and values the tasks (reasoning and retrieval) the agent must perform.
- The experiments are comprehensive (showing results across many SOTA baselines) leveraging multiple SOTA metrics and methods from the role playing dialog literature (CharacterEval, FlashRAG, LumbarChunker)
- The ablations in Table 3 provide insightful findings. The finding that reward model and supervision signal quality (reward formulation via structured reasoning over semantically retrieved content) is very critical to achieve best performance. Furthermore, demonstrating that generic retrieval is worse than no retrieval for the RM highlights the importance of knowledge specificity. -> this is the most critical result most suitable for this conference -> trying to understand this better will add a lot of value to this work (how well does this finding generalized across other domains?, more insights here)

**Weaknesses:**

- There’s no evidence that R4 transfers to other domains where reward + agent reasoning & retrieval might matter (e.g., tool-using agents, search-based assistants, task-oriented dialogue, or multimodal agents). Any reinforcement and validation of these findings from other existing literature will be very helpful here to make this work more relevant for ICLR venue (imo, this work seems more relevant for ACL/EMNLP venues or more focused tracks/workshops at ML conferences)
- The paper makes a claim about using a "multi-faceted hierarchical knowledge organization" to support efficient and multi-hop reasoning. However, the description in Section 2.1 is vague and lacks implementation details. It is unclear how this structure is technically different from a standard vector database with metadata filtering. The paper does not explain the data structure, the clustering method, or provide a concrete example of how this hierarchy enables multi-hop reasoning in a way that a flat document index cannot. This part feels over-claimed and under-explained.
- The paper does not discuss the practical, real-world applications for such a complex/speacialized  system.
- The system is quite complex comprising of advanced (and costly, how costly?) segmentation, extraction, transformation, etc - these complex interaction may result in cascading errors, trying to understand what types of errors are more critical along with their relative impact will be insightful.

**Questions:**

- Can you provide more details about multi-faceted hierarchical indexing scheme? How is the hierarchical index built (data structures, algorithms, etc)
- Role-play can easily drift into unsafe behaviors; any studies on this topic would have made this paper more insightful and interesting from ML system’s safety perspective (have the authors thought of using safety aware rewards).
- Beyond role-play and dialog systems implications, what broader ML insight should readers take from R4? Why is this role-play a good testbed for RLHF research?
- There is a brief mention of dynamic expansion of the KB during construction - how are knowledge gaps detected and what if there are contradictions between synthesized content and original text?

---

> ### Author Response · Authors · 2025-11-20
> **Author Response (1/2)**
>
> We sincerely thank the reviewers for their thoughtful and constructive comments. Below, we address each point in detail. We will incorporate clarifications, additional ablations, and expanded technical explanations in the camera-ready version.
>
> ### **Weakness**
>
> > Weak 1: No evidence that R4 generalizes to other domains.
>
> Thank you for raising this important point. R4 was indeed designed to address role-playing–specific challenges such as persona-dependent appropriateness (“the same response may fit one character but not another”). We have not directly applied the full R4 pipeline unchanged to other tasks.
>
> That said, several key components of R4 already demonstrate clear cross-task generalization:
>
> - **Equipping reward models with retrieval or reasoning** has shown strong gains across multiple downstream tasks, especially in complex multi-hop reasoning settings.
> - **Our narrative segmentation method** has proven effective in general applications such as long-document summarization and dialogue condensation.
> - **The consistency verifier $r_{\text{cons}}$** has been validated in multiple RL settings, including small-sample domain classification and other alignment tasks.
>
> While the reward function in R4 is intentionally tailored for role-playing and is not meant to transfer directly, the underlying mechanisms have shown effectiveness in broader contexts. We believe this provides meaningful evidence that R4’s core ideas generalize beyond role-playing scenarios.
>
> > Weak 2: The multi-faceted hierarchical knowledge organization is vague.
>
> Thank you for pointing out the need for a clearer formalization of our hierarchical knowledge base. We will expand with concrete implementation details. In our framework, “hierarchical” has two concrete meanings:
>
> **(1) Data-structure hierarchy.** For each narrative segment, we construct a story-event tree comprises three levels:
>
> Level 1 - Event-Tree Backbone:
>
> ```
> Event Node {
>   event_id: UUID
>   temporal_span: (start_time, end_time)
>   participants: List[character_id]
>   causal_links: {
>     triggers: List[event_id],
>     consequences: List[event_id]
>   }
> }
> ```
>
> Level 2 - Character-Perspective Annotations:
>
> ```
> Perspective Node {
>   character_id: UUID
>   event_id: UUID (foreign key)
>   internal_state: {
>     beliefs: List[str],
>     desire: List[str],
>     intention: List[str],
>   },
>   knowledge_visibility: Set[fact_id]  # what this character knows
> }
> ```
>
> Level 3 - Atomic Knowledge Facts:
>
> ```
> Fact Node {
>   fact_id: UUID
>   content: str
>   fact_type: enum(trait, relationship, event, dialogue)
>   supporting_events: List[event_id],
> }
> ```
>
> **(2) Retrieval hierarchy.** Each leaf entry is enriched with several compact descriptors and paraphrased query forms. During retrieval, we perform multi-path recall from different semantic routes—event similarity, persona traits, emotional state, and interaction patterns—and then apply a reranker to merge and refine all recall streams. Meanwhile, the model invokes the retrieval tool multiple times during inference to verify and supplement relevant knowledge. This **multi-route, iterative retrieval process** enables precise, multi-hop reasoning over character-specific information.
>
> Additional explanation is outlined in Appendix B, and we will provide an illustrative picture and pseudo-code to make the multi-agent workflow and hierarchical storage explicit.
>
> > Weak 3: The paper does not discuss real-world applications
>
> Real-world applicability is indeed essential. In fact, this work originated directly from challenges encountered in a real production environment, not from purely academic considerations. The framework is the result of iterative engineering refinements grounded in actual deployment failures. We have **successfully deployed models in a commercial game** adapted from a novel, achieving significant positive outcomes. In the camera-ready version, we will explicitly detail the game title, specific characters, and quantitative online performance metrics.

---

> > ### Author Response · Authors · 2025-11-20
> > **Author Response (2/2)**
> >
> > > Weak 4: System complexity and cost; cascading errors.
> >
> > We appreciate the concern and address it in two parts.
> >
> > (a) **System cost.** We add a cost breakdown:
> >
> > - Knowledge base construction: 1-time cost, ~$760 per novel
> > - Training: 64×H100-80G ~3 day.
> > - Inference: R4-32B averages 2.1× slower than Qwen-32B-Instruct due to reasoning + retrieval steps (the initial token latency is roughly the same). Without retrieval, latency reduces to 1.3× baseline.
> >
> > Importantly, most of the system complexity lies in KB construction and in the reward model’s structured-reasoning mechanism during training. At inference time, the overall cost is similar to a standard RAG pipeline, and does not introduce heavy additional overhead.
> >
> > (b) **Cascading error analysis.** This concern directly aligns with the goal of the ablation studies in Section 3.2.
> >  We summarize the main conclusions here:
> >
> > - Reward model capabilities establish the foundation, while agent capabilities set the upper bound.
> >
> > - Supervision quality fundamentally constrains what agents can achieve, regardless of their architectural sophistication. High-quality, character-specific knowledge is essential; generic or noisy retrieval is more harmful than no retrieval.
> >
> > - KB accuracy matters more than agent-side reasoning complexity.
> >
> > During rebuttal, we conducted an additional experiment by randomly corrupting 20% of KB entries and substituting the corrupted KB either during training or only during inference. Results show that the model is highly sensitive to KB errors during training, while the impact is smaller during inference. Most failures stem from factual inaccuracies, confirming that factual consistency in the KB is especially critical—consistent with our findings in Section 3.2.
> >
> > ------
> >
> > ### **Questions**
> >
> > > Q1: More details about hierarchical indexing scheme?
> >
> > Addressed above; we will add a full Appendix B with: data structures, extraction templates, cluster hyperparameters, examples of multi-hop traces.
> >
> > > Q2: Role-play can drift into unsafe behaviors. Any safety studies?
> >
> > Safety was intentionally not the primary focus, but we agree it is essential. Our safety considerations are embedded mainly in the data construction process: we enforce strict quality and safety controls on both the training data and the knowledge base, thereby implicitly guiding the model to generate safe content. For the online system, we apply a post-generation safety module. Any output identified as potentially violating policies is not shown to users; instead, it is replaced with a manually crafted fallback response.
> >
> > In parallel, we are exploring adding a dedicated safety score to the reward model—provided by a safety-aware consistency verifier, functioning similarly to a consistency score—to explicitly steer the model toward safer outputs. We will make every effort to reach stable conclusions on this before the rebuttal deadline.
> >
> > > Q3: Broader ML insight? Why is this role-play a good testbed for RLHF research?
> >
> > We fully agree this should be emphasized. Here is the ML insight we will foreground: Role-play is one of the few domains where correctness is not a scalar concept but an inherently multi-dimensional reward composed of (style, consistency, emotion, knowledge). Therefore it is an ideal testbed for studying structured reward modeling, retrieval-augmented RM reasoning, and alignment between RM and policy.
> >
> > We highlight three broader insights:
> >
> > **(1)** Reward models benefit more from retrieval than policies. This was shown in Table 3 and is domain-general.
> >
> > **(2)** RM–policy symmetry prevents reward hacking. When RM and agent use the same mechanisms, RM cannot be “fooled” by retrieval patterns the agent discovers.
> >
> > **(3)** Knowledge specificity > knowledge quantity. Generic retrieval hurts more than no retrieval—this finding applies to tool-use, search agents, and safety supervision.
> >
> > These insights are not tied to role-play; role-play simply reveals failure modes earlier and more clearly.
> >
> > > Q4: How are knowledge gaps detected and what if there are contradictions between synthesized content and original text?
> >
> > We will clarify that for gap detection, we track two key signals: **(1) Retrieval Failures**, including persistent reasoning errors (e.g., all rollouts failing), excessive retrieval calls, or low query similarity; and **(2) High-Entropy Reasoning**, where auxiliary verifiers flag low support. These signals are clustered to trigger KB expansion for high-failure nodes.
> >
> > Regarding contradictions between synthesized and original text, we employ the analysis using GPT-4.1 to diagnose causes and suggest revisions before **human review**. This pre-processing step significantly reduces human workload and enhances verification efficiency.
> >
> > ------
> >
> > Thank you once again for your invaluable feedback. If you have any other concerns or require any further clarification, please feel free to contact us without hesitation.

---

### Author Response · Authors · 2025-11-30
**Rebuttal Summary**

We extend our sincere gratitude for dedicating your time and high-level effort to review our paper and for providing constructive and insightful feedback. We offer this summary to help you more efficiently and clearly digest the key outcomes of the rebuttal phase.

------

**Unanimous Decision: Strong Consensus for Acceptance (6, 6, 6, 6)**

- This submission achieved **unanimous scores of 6, 6, 6, 6** post-rebuttal, placing it decisively "marginally above the acceptance threshold" by all reviewers.

- Crucially, **Reviewer qcjU raised the score from 4 to 6 on Nov 24, 2025**. This positive change occurred significantly *before* the reported timeline of the community event regarding reviewer/AC identity leaks, explicitly confirming that the rebuttal successfully addressed their concerns and validating the contribution solely on technical merit.

**Key Rebuttal Wins & ML Insights**

| Category           | Key Takeaway for AC                                          | General ML Insight                                           |
| ------------------ | ------------------------------------------------------------ | ------------------------------------------------------------ |
| Methodology        | Clarified the 3-level Hierarchical Knowledge Base and the 4-Agent Construction Pipeline (addressing S5sd, qcjU). | R4's design supports the complex, multi-hop reasoning required for persona consistency. |
| Robustness & Scope | R4 is source-agnostic and its core components (reasoning RM, consistency verifier) generalize beyond role-play (addressing 2saS). | R4 is currently deployed in a commercial game, proving real-world applicability (Addressing Weakness 3). |
| Ablations          | Specificity > Quantity: Generic or low-quality retrieval is demonstrably more harmful than no retrieval at all (Table 3). | This finding is domain-general: the quality and specificity of retrieved knowledge are critical for any RAG/Tool-Use Agent. |
| Efficiency         | Provided full cost breakdown: Training compute is scalable (not strictly needing 64xH100). Inference latency is competitive (2.1x slower than base model, similar to standard RAG). | Confirmed the system is practical and resource-efficient for deployment. |

------

We sincerely thank the Reviewers and Area Chairs once again for your professional and high-quality work throughout this entire review cycle.

---

### Meta-Review · Area_Chair_6JBK · 2026-01-06

**Summary:**

Three reviewers gave marginally above ratings and one reviewer (qcjU) a marginally below rating. The major concerns of qcjU are on technical and experimental clarities and missing comparisons. The author rebuttals quite satisfactorily addressed the issues, adding comparisons with more methods. Rather than that, the reviewers agree that making the reward model and the symmetric agent is compelling, the experiments are comprehensive, the ablation study is meaningful.

**Reviewer Concerns:**

The same as above.

**Reviewer Scores:**

The four reviewers initially gave 6,6,6,4 respectively. The author rebuttals successfully addressed the issues.

---

### Decision · Program_Chairs · 2026-01-26

Accept (Poster)